# Modeling and application of production metering for electric pump wells without downhole pressure measurement devices

Yongcang Ren[1,2,3,4], Da Guo[1,2,3,4], Yizhen Sun 📙[5*], Bao Zhang[1,2,3,4], Jianxin Shen[1,2,3,4], Yingbin Liu[1,2,3,4], Yinglin Zhang[1,2,3,4], Guoxiang Zhang[1,2,3,4], Jinsong Yao[5], Xueling Du[1,2,3,4]

1 R&D Center for Ultra Deep Complex Reservoir Exploration and Development, CNPC Korla, Korla, Xinjiang, China, 2 Engineering Research Center for Ultra-deep Complex Reservoir Exploration and Development, Xinjiang Uygur Autonomous Region Korla, Korla, Xinjiang, China, 3 Xinjiang Key Laboratory of Ultra-deep Oil and Gas Korla, Korla, Xinjiang, China, 4 PetroChina Tarim Oilfield Branch, Korla, Xinjiang, China, 5 Research Institute of Petroleum Exploration and Development, Beijing, China

* 605182894@qq.com

## Abstract

To address the issues of low frequency and high costs associated with the current manual production measurement for ESP wells in the Tarim Oilfield, a study was conducted to develop a digital production measurement method for ESP wells. Based on the principle of energy conservation, where the input power of the pump equals the output power of the motor, and incorporating parameters such as surface tubing and casing pressure, motor current, and motor/ pump performance curves, with viscosity correction of the pump performance curve, a corrected power calculation method was proposed. A digital production measurement mathematical model was established. According to feedback from field applications, the calculated results of this method align well with the metered results when corrected using on-site measured flow rate. Furthermore, by applying this model, accurate allocation of merged production well outputs and risk warning or failure diagnosis for oil wells can be achieved. This method not only improves the accuracy and efficiency of ESP well production calculations but also enables real-time reflection of oil well production trends, contributing to intelligent production management in the Tarim Oilfield and significantly enhancing the level of oilfield production management.

## 1 Introduction

The Tarim Oilfield currently operates 1,288 artificial lift wells, including 336 electric submersible pump (ESP) wells, with a daily oil production of 1,980.62 tons. As one of the most crucial artificial lift methods in Tarim, ESP wells play a key role in oilfield development. However, the current production calculation methods primarily rely on manual measurements, which involve high labor intensity, low measurement

**Data availability statement:** All relevant data are within the manuscript and its Supporting Information files.

**Funding:** The author(s) received no specific funding for this work.

**Competing interests:** The authors have declared that no competing interests exist.

frequency, and significant errors caused by sampling biases from personnel. The lack of automation severely impacts the management efficiency and production performance of ESP wells [1–6]. Compared with conventional flowmeter-based measurement methods, establishing computational models to predict flow rates offers advantages such as higher measurement frequency and lower costs. This approach can effectively capture the dynamic variations of oil wells and facilitate the analysis and evaluation of the oilfield's production characteristics.

At present, the digital production measurement methods for electrical submersible pumps (ESP) primarily include the pressure differential method at the pump inlet and outlet, the motor-ESP energy balance method, and the artificial neural network (ANN) model method. Each of these approaches has its own advantages, but they also face challenges in practical applications. The pressure differential method calculates the actual head generated by the ESP using the pressure difference between the pump inlet and outlet, combined with the fluid density inside the pump. By referencing a corrected ESP head curve, production rates can be obtained. While this method is straightforward, it is only applicable to wells equipped with downhole sensors capable of simultaneously measuring the pressures at both the pump outlet and inlet [7].The motor-ESP energy balance method establishes a relationship between well flow rates and electrical parameters based on the principle of energy conservation between the motor and the pump, enabling production measurement. However, this method lacks a correction mechanism for the ESP characteristic curve, limiting its accuracy and applicability [8–12]. To address this, researchers have proposed several improvements, such as integrating viscosity correction models for ESP characteristic curves [13], employing multiphase flow models to calculate pump outlet pressure, and introducing flow correction coefficients [14,15]. Despite these advancements, the improved models often require downhole sensors or capillary pressure measurement devices. Additionally, the empirical formulas used for calculating motor power factors and efficiency are typically specific to certain pump types, which may lead to errors when applied to other ESPs in real-world operations [16,17].Another approach involves using ANN models or other machine learning algorithms to establish production prediction models for ESP wells. Although these models are relatively simple in principle and offer low production measurement errors, they require substantial field data for training and heavily depend on the reliability of the data [18].

This study focuses on the characteristics of electrical submersible pump (ESP) wells in the Tarim Oilfield and optimizes the traditional energy balance method to develop an online production measurement model for ESP wells. The application of this model enables production measurement for ESP wells, allocation of production from commingled metering wells, and casing damage early warning for ESP wells. It holds significant importance for reducing costs, improving efficiency, and enhancing the level of intelligence in oilfield operations.

## 1.1 Establishment of the mathematical model

The production system of electrical submersible pump (ESP) wells uses electrical energy as the power source. The surface power is transmitted to the downhole motor

through the submersible cable, driving the multi-stage centrifugal pump to rotate at high speed. This process converts electrical energy into mechanical energy, providing sufficient energy to lift the fluid through the tubing to the surface. Assuming the ESP well production system operates under steady-state conditions, the torque of the pump and the speed of the motor remain stable [19]. Based on the principle of energy conservation, the input power of the ESP is equal to the output power of the motor, which can be expressed as:

$$P_{\text{pump}} = P_{\text{motor}} - \Delta P \tag{1}$$

In the formula, $P_{\text{pump}}$ represents the input power of the pump, kW; $P_{\text{motor}}$ represents the output power of the motor, kW; $\Delta P$ is the correction power (kW), representing the energy consumption caused by the protector and intake or gas separator, as well as the energy loss due to equipment aging and wearing.

The input power of the pump is given by the following formula:

$$P_{pump} = 11.574 \frac{\Delta p Q_{\text{p}}}{\eta_{\text{p}}} \tag{2}$$

In the formula, $\Delta P$ represents the pump pressure difference,MPa; $Q_P$ represents the flow rate of the electric submersible pump,m³/d; $\eta_P$ represents the pump efficiency,dimensionless. Here, 11.574 represents the unit conversion factor.

In the electric submersible pump production system, the pump unit can be regarded as a series circuit, meaning the current metered on the surface is equal to the input current of the motor. The output power of the motor is derived from the motor characteristic curve, which is determined by the motor current and operating frequency. The formula for the motor output power is as follows:

$$P_{motor} = P'_{\text{motor}} \times \frac{f_0}{f_{\text{np}}} \times \beta \tag{3}$$

In the equation, $P'_{motor}$ is the rated output power of the motor at the rated frequency, kW; $f_o/f_{np}$ represent the actual operating frequency and the rated frequency of the motor, Hz; $\beta$ is the motor load factor, dimensionless.

In the equation, the selection of parameters is as follows:

(1) The pump inlet and outlet pressure difference $\Delta P$ is the difference between the pump outlet pressure and the pump inlet pressure [20]. Since the Tarim oil field electric pump wells do not have an underground pressure measurement device, other methods need to be employed for calculation. First, using historical dynamic fluid levels and casing pressure data, the pressure at the well bottom perforation zone is calculated. By combining the corresponding production data and static fluid level or reservoir static pressure data, the inflow performance relationship (IPR) curve is obtained for the oil well [21]. Then, using the Beggs-Brill tubing multiphase flow model [22,23], the wellbore pressure at the pump suction inlet is calculated starting from the bottom flow pressure, allowing the intake inflow performance relationship (IIPR) [22]curve for different production rates to be derived. Next, starting from the wellhead pressure, the multiphase flow model is used to calculate the pressure at the pump outlet, obtaining the vertical lift performance (VLP) curve [24] for different production rates. By subtracting the IIPR curve from the VLP curve, the pump inlet-outlet pressure difference versus production relationship curve is obtained;

(2) The pump efficiency $\eta_P$ can be obtained from the electric pump characteristic curve. However, it is necessary to consider the effects of parameters such as viscosity, and to correct the electric pump characteristic curve accordingly. This results in a revised production-efficiency curve;

(3) The actual operating frequency of the motor$f_o$ can be directly obtained from the frequency converter settings, while the rated frequency of the motor$f_{np}$ can be obtained from the motor nameplate information. Since the rated output power

of the motor is typically metered at its rated frequency, the actual power output may vary when the motor operates under variable frequency conditions. Therefore, it is necessary to consider the impact of frequency on both the power output and operational stability of the motor;

(4) The motor load $\beta$ can be determined based on the ground current value metered on-site, combined with the motor characteristic curve.

Based on the above formula, an energy balance model for production calculation is established. The calculation process is shown in Fig 1.

The steps are as follows:

1. Correct the electric pump's factory characteristic curve based on the pump's operating frequency and the average fluid viscosity, to obtain the discharge-efficiency characteristic curve of the electric pump under actual operating conditions.

2. Calculate the wellbore pump intake and discharge dynamic curves, then determine the pump inlet and outlet pressure differences at different production rates. Combine this with the electric pump's discharge-efficiency characteristic curve to obtain the electric pump input power required for the well at different production rates.

3. Based on the on-site metered surface operating current and the motor characteristic curve, calculate the actual load factor of the motor. Then, use this load factor and the motor's rated output power at the actual operating frequency to calculate the motor's actual output power under the current operating conditions.

4. Using the energy balance in the electric pump production process, where the pump input energy equals the motor output energy, use the calculated motor output power and the corrected discharge-power curve to obtain the final production result.

## 2 Model parameters calculation

### 2.1 Pump inlet pressure and pump outlet pressure

When calculating the input power of the electric pump, the inlet and outlet pressures of the pump are required.

For an electric submersible pump well, the fluid enters the pump through the suction inlet, and the fluid in the annular space above the pump suction inlet remains stationary. After a stable production period, due to the difference in oil-water

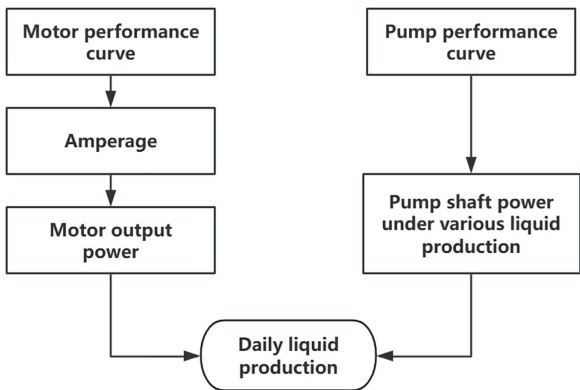

**Fig 1. Flowchart of the production calculation method.**

density, the fluid above the pump is entirely oil [25]. Therefore, the pressure generated by the liquid column above the pump can be regarded as the suction inlet pressure of the pump, and the calculation formula is as follows:

$$P_\text{i} = \rho_o g h_\text{flop} \tag{4}$$

In the formula, $p_i$ is the pump inlet pressure, MPa; $\rho_0$ is the crude oil density, kg/m³; g is the gravitational acceleration, m3/s; $h_{fop}$ is the vertical height difference between the wellbore liquid level and the pump suction inlet, m.

By substituting the value $h_{fop}$ of dynamic liquid level and static liquid level into the above formula, the pump inlet pressure corresponding to different production rates can be obtained.

Below the pump suction inlet, the fluid flows through the casing, from the perforated section to the electric pump suction inlet. This section requires the use of the Beggs-Brill multiphase flow model for wellbore calculations. The pump inlet pressure calculated from the previous step is used as the starting point, and the flow pressure at the bottom of the well is calculated to the perforated section.

The reservoir static pressure is set as the intercept. Using the flow pressure at the bottom of the well for different production rates, the slope b is fitted using the least squares method, and the regression equation is established to obtain the Inflow Performance Relationship (IPR) curve for the well:

$$P_\text{wf} = b Q_\text{P} + p_r \tag{5}$$

In the equation, $p_{wf}$ represents the bottomhole flow pressure, MPa, Q is the production rate, m³/d, $p_r$ is the reservoir static pressure, MPa。

After that, based on the oil well's IPR curve, the wellbore multiphase flow model is used to calculate the flow to the pump suction inlet, obtaining the oil well's Intake Inflow Performance Relationship (IIPR) curve Fig 2.

For the pump outlet pressure, the wellhead tubing pressure can be used as the starting point. Using the pump outlet position as the solving node, the Beggs-Brill wellbore multiphase flow model is applied to calculate the pump outlet pressure. By calculating the pump outlet pressure corresponding to different liquid production rates, the wellbore's outflow performance curve is obtained Fig 3.

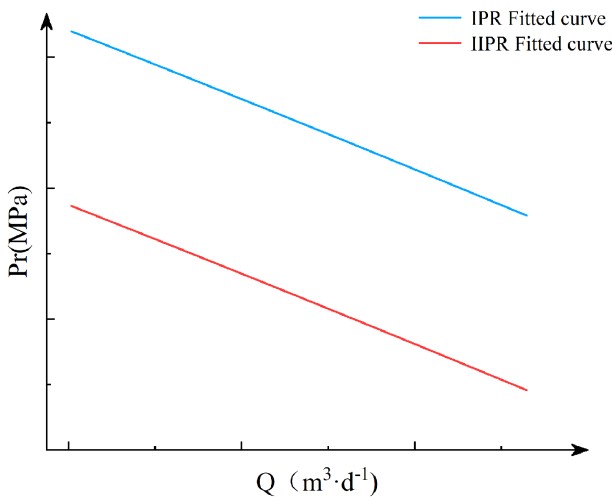

**Fig 2. Fitting of IPR and IIPR curves.**

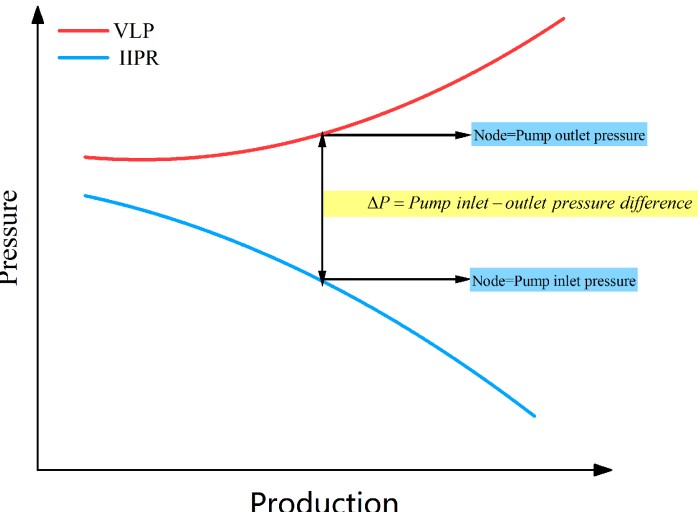

**Fig 3. Calculate the pressure difference at the pump inlet and outlet.**

## 2.2 Correction of the electric pump flow-efficiency curve

First, the influence of speed on the pump's characteristic parameters is considered. Since the given electric pump characteristic curve is metered at the rated pump speed, the electric pump flow rate must be corrected using the pump affinity law to obtain the flow-efficiency curve at the given operating frequency.

Next, the influence of fluid viscosity on the characteristic curve is considered. Subsequently, the influence of fluid viscosity on the characteristic curve is considered. According to Equation (2), pump efficiency has a significant impact on the calculation of the pump input power. Based on the research by Luiz Pastre et al [26], the best efficiency point (BEP) of a centrifugal pump is highly affected by fluid viscosity. This is because high-viscosity fluids significantly increase frictional losses during flow within the pump, leading to reduced pump efficiency and deviations in the head-flow relationship from the characteristic curve under clean water conditions. Through viscosity correction, the actual performance of the pump under high-viscosity fluids can be more accurately reflected, optimizing the relationship between head, flow rate, and efficiency, thereby improving the accuracy of power calculations.

In this model, viscosity correction is applied using the centrifugal pump characteristic curve viscosity correction method provided by the ANSI standard [26] to correct the electric pump efficiency characteristic curve. The correction steps are as follows Fig 4:

1. Calculate the performance parameter B, with the formula:

$$B = 16.5 * \frac{(V_{vis})^{0.50} \times (H_{BEP-W})^{0.0625}}{(Q_{BEP-W})^{0.375} \times N^{0.25}}$$

(6)

In the formula, $V_{vis}$ represents the dynamic viscosity, cSt; $H_{BEP-W}$ is the head corresponding to the best pump efficiency point under water conditions, m; $Q_{BEP-W}$ s the flow rate corresponding to the best pump efficiency point under water conditions, m³/h; $N$ is the rotational speed, RPM。

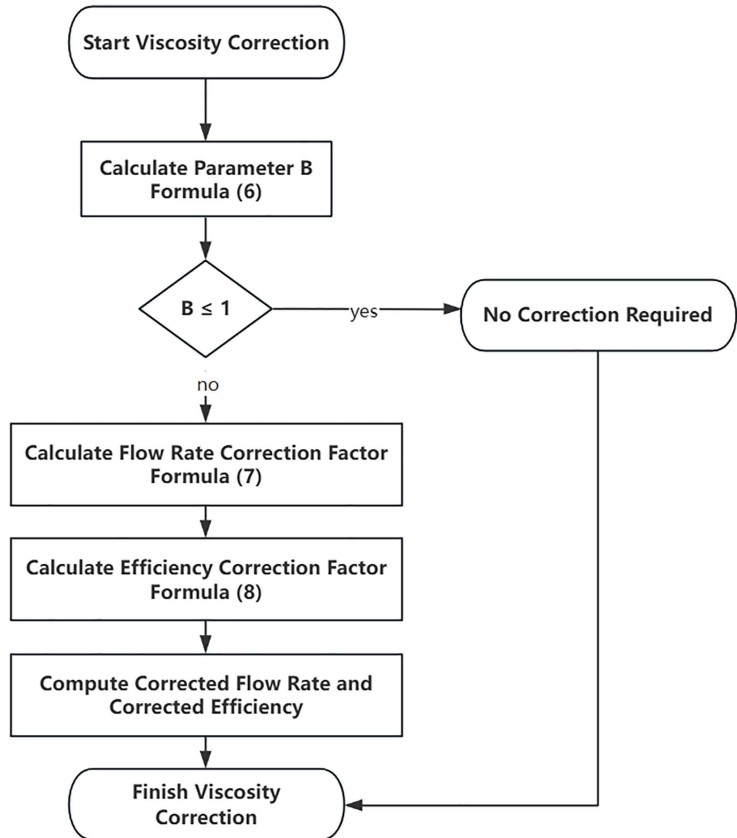

**Fig 4. Viscosity Correction Flowchart.**

If B ≤ 1, no correction is needed; otherwise, the following steps should be performed:

2. Calculate the corrected flow rate. The flow rate correction factor is given by the following formula:

$$C_Q = (2.71)^{-0.165 \times (\log_{10} B)^{3.15}}$$

(7)

Multiply each flow rate by the flow rate correction factor to obtain the corrected flow rate.

3. Calculate the pump efficiency correction factor $C_\eta$. The calculation method is as follows:

$$C_\eta = B^{-(0.0547 \times B^{0.69})}$$

(8)

Multiply each efficiency by the efficiency correction factor to obtain the electric pump efficiency under the specific viscosity fluid condition.

Using the calculated equivalent flow rate and equivalent efficiency, a new pump performance curve is plotted, which represents the viscosity-corrected pump production-efficiency curve.

## 2.3 Calculation of motor output power

During the actual operation of the motor, various factors, such as component aging and load changes, can affect its performance, causing deviations from the factory characteristic curve. Therefore, it is necessary to measure the ground

current, find the corresponding load factor on the motor characteristic curve, and calculate the motor's output power using formula (3).

## 2.4 Corrective power

In actual production, the shaft power output by the submersible electric motor must be transmitted to the submersible electric pump through protectors, intake devices, separators, and other components [27]. Therefore, some energy is consumed. Additionally, for electric pump units that have been in operation in oil wells for a period of time, due to issues such as pump wear, sediment accumulation, and motor component aging, the actual performance curve of the electric pump and motor may deviate from the theoretical values calculated based on the factory characteristic curve, leading to errors in the calculation of electric pump and motor power.

To improve the accuracy of the calculations, the actual metered production data and ground current data can be used. The difference between the calculated electric pump input power and motor output power is taken as the correction power, which is then substituted into the model to solve for the production rate.

## 3 Model solution

According to the established production calculation model, it is necessary to obtain the relationship curve between the electric pump input power and the production rate, and solve for the production rate corresponding to the electric pump input power equal to the motor's actual output power.

It is important to note that the electric pump power curve needs to be drawn based on the fluid flow rate inside the pump, whereas the IIPR and VLP calculations use the surface liquid production rate. Considering the compressibility of the reservoir-produced fluid and the solubility of natural gas, the fluid flow rate inside the pump and the surface liquid production rate may not be the same. Therefore, the corresponding relationship between the surface liquid production rate and the fluid flow rate inside the pump needs to be considered during the model calculation process. This step requires calculating the gas/liquid phase fluid volume flow rate under the pump inlet and outlet pressure conditions, based on the fluid properties, and then taking the average.

The model solving process is as follows:

1. Based on historical dynamic liquid levels and production data, calculate the corresponding well bottom flow pressure, and combine with reservoir static pressure data. Use the least squares method to fit and obtain the well's IPR (Inflow Performance Relationship) curve. Then, use the Beggs-Brill tubing multiphase flow calculation model to calculate the pump inlet pressure corresponding to different production rates, thus obtaining the well's IIPR (Intake Inflow Performance Relationship) curve.

2. Combine the IIPR curve, assume a set of surface liquid production rates, and use the Beggs-Brill tubing multiphase flow calculation model. Starting with the wellhead oil pressure, calculate the pump outlet pressure under different surface liquid production conditions, thus obtaining the well's VLP (Vertical Lift Performance) curve.

3. For each surface production rate, calculate the corresponding pump inlet and outlet pressures, and use the fluid property calculation model to calculate the gas/liquid phase volume flow rate under the pump inlet and outlet pressure conditions. Then, calculate the average volumetric flow rate of the fluid inside the pump, which represents the pump's displacement. After this step, the corresponding relationship between surface liquid production and pump internal displacement is obtained.

4. Use the electric pump operating frequency and pump internal fluid viscosity data to correct the factory efficiency characteristic curve of the electric pump, obtaining the pump's displacement-efficiency characteristic curve under actual operating conditions.

5. Use the IIPR curve and VLP curve to calculate the pressure difference at the pump inlet and outlet for each surface liquid production rate. Use the results from step 3 to calculate the pump internal fluid displacement corresponding to each surface liquid production rate. Use the results from step 4 to calculate the pump efficiency for each surface liquid production rate. Substitute these parameters into formula (2) to calculate the required electric pump input power corresponding to each surface liquid production rate, and perform interpolation to obtain the surface liquid production rate-electric pump required input power curve.

6. Use actual metered liquid production data and substitute into the previous step to calculate the corresponding electric pump actual input power. Use the surface current and motor characteristic curve to calculate the motor's actual load rate, and combine with the motor's rated output power to calculate the motor's actual output power. The difference between the motor output power and the electric pump actual input power is the correction power.

7. Use the metered surface current to solve for the motor output power. The difference between this power and the correction power is the electric pump's actual input power. Combine the surface liquid production rate-electric pump required input power curve to solve for the surface liquid production rate corresponding to the electric pump actual input power. This is the calculated production rate.

## 4 Field applications

### 4.1 Model production calculation results

To verify the feasibility of this production calculation method, a validation was conducted on multiple electric submersible pump (ESP) wells in the Tarim Oilfield, using Well H17 as an example. The well depth is 2500.4 meters, and the motor's rated voltage, current, and output power at 50 Hz are 1173 V, 40 A, and 56 kW, respectively. The electric pump operates at a frequency of 50 Hz. The characteristic curves of the electric pump and motor used in this well are shown in the figure below Figs 5 and 6.

First, calculate the VLP and IIPR curves for this well, as shown in the figure below Fig 7.

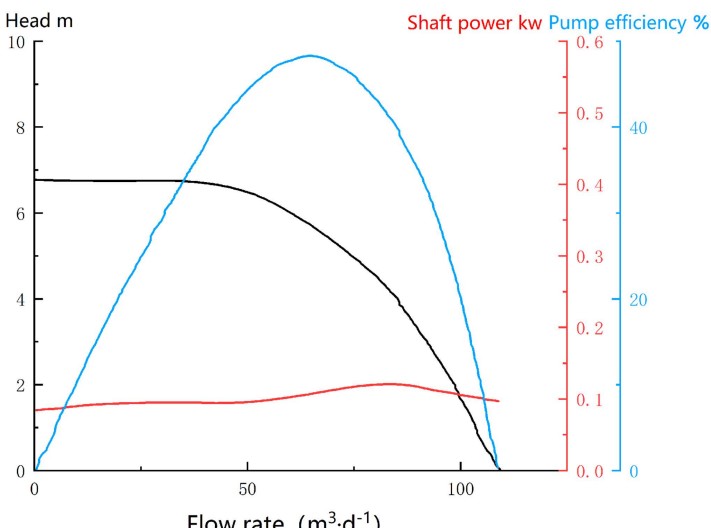

**Fig 5. Characteristic Curve of the Electric Submersible Pump in Well H17.**

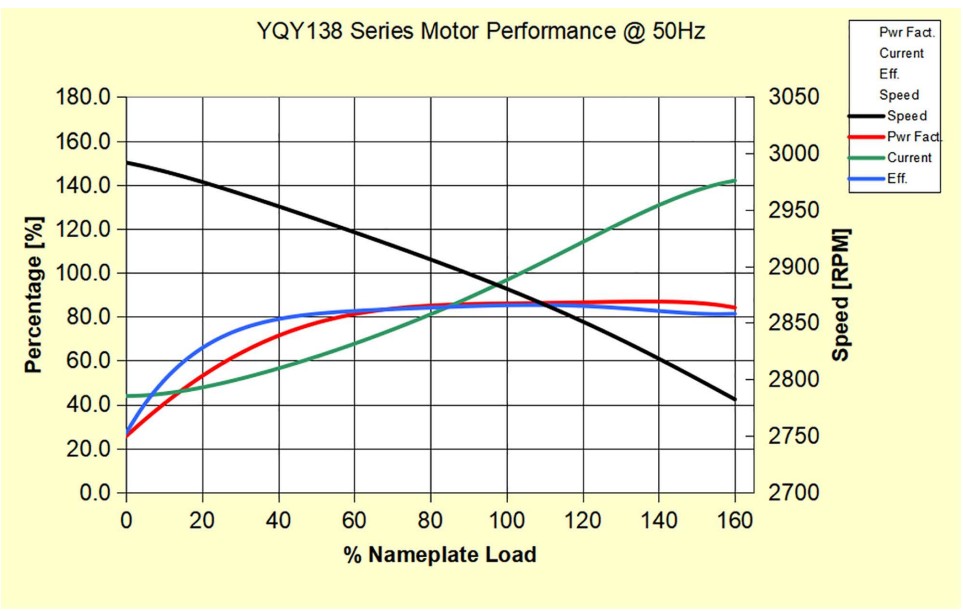

**Fig 6. Motor Characteristic Curve for Well H17.**

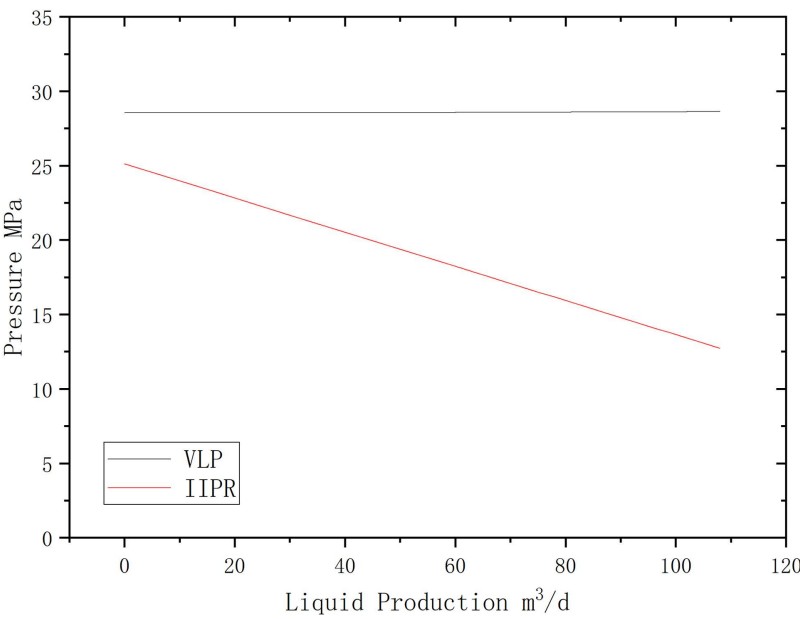

**Fig 7. VLP and IIPR Curves for Well H17.**

Next, correct the electric pump efficiency characteristic curve. The comparison of the efficiency curves before and after correction is shown in the figure below Fig 8.

According to formula (2), calculate the electric pump input power required for different surface liquid production rates, as shown in the figure below Fig 9.

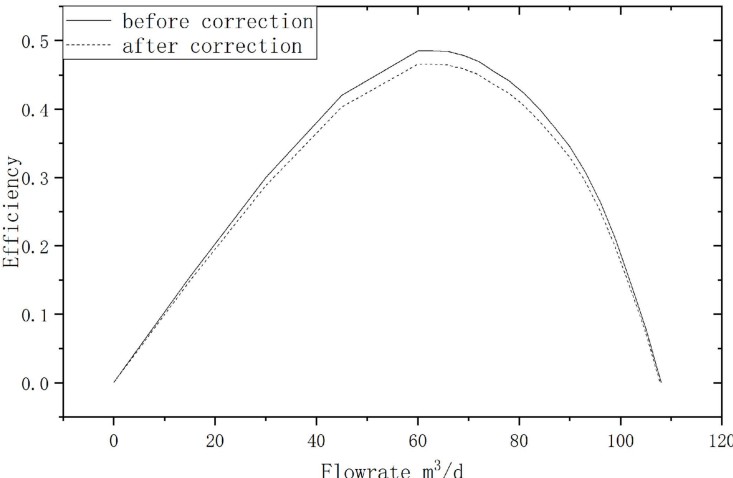

**Fig 8. Comparison of Electric Submersible Pump Characteristic Curves Before and After Correction.**

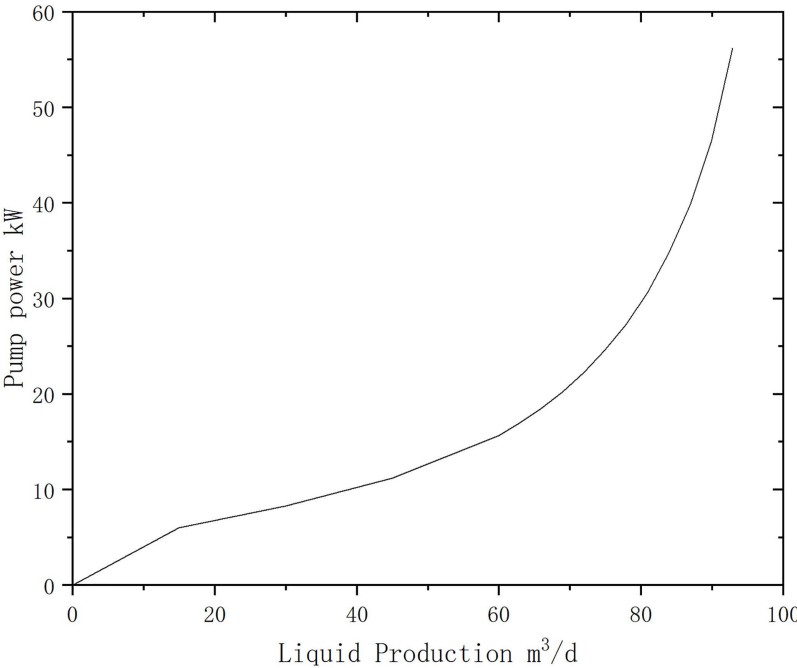

**Fig 9. Input Power Required for the Electric Submersible Pump.**

Using the data from December 13, 2023, for this well, the efficiency correction was calculated. On this day, the surface liquid production rate was 84 m³/d, and the average daily current was 29.7 A. Based on the actual surface liquid production rate, the required electric pump input power was calculated to be 34.5 kW. Using the average daily current and the motor characteristic curve, the motor load was found to be 0.698. Combining this with the motor's rated output power of 56 kW at 50 Hz, the actual motor output power was calculated to be 39.1 kW. Therefore, the correction power is the difference between the actual motor output power and the electric pump consumption power, which is 4.6 kW.

From December 13, 2023, to April 23, 2024, the actual metered daily liquid production, average daily current, and model-calculated daily liquid production data are shown in the figure below. According to the calculations, the average error between the model-calculated daily liquid production and the actual metered production during this period was about 4.74% Fig 10.

### 4.2 Calculation of production allocation for multiple combined metered wells

Due to issues such as surface pipeline failures or insufficient personnel, it is sometimes necessary to combine the production measurements of multiple wells. Therefore, the total metered production needs to be allocated to determine the production of each individual well.

Assuming a metering station measures the total liquid production from N wells, with the flowmeter at the metering station measuring a real-time production rate of $Q_m$, the total liquid production $Q_c$ for all wells at the same metering station is calculated using the previously described method for measuring liquid production:

$$Q_c = \sum_{i=1}^{N} Q_{ci}$$

(9)

Where: $Q_{ci}$ is the calculated production of well i on the platform, m³/d; N is the total number of wells included.

The total production error can be obtained by subtracting the calculated total production from the metered total production:

$$\Delta Q = Q_m - Q_c$$

(10)

When calculating the single-well production rate, the error between the calculated production and the metered production can be distributed to each well according to a specific weight factor to correct the production of each well.

Set the weight factor for each well as follows:

$$\omega_i = \frac{Q_{ci}}{\sum_{i=1}^{N} Q_{ci}}$$

(11)

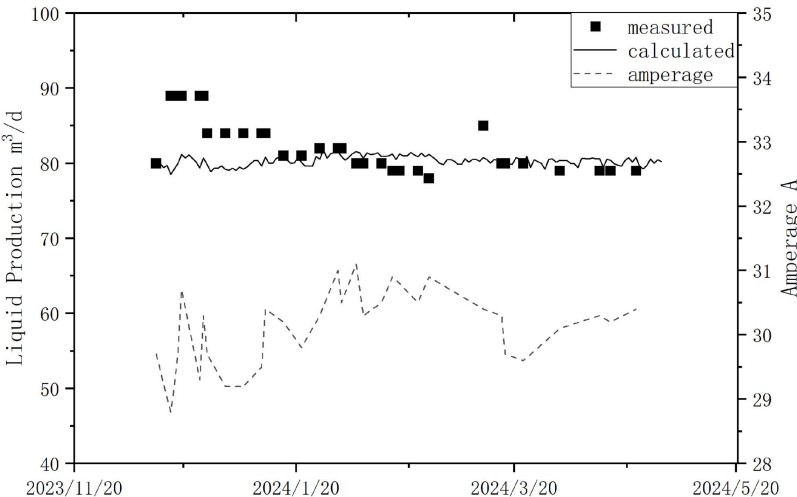

**Fig 10. Daily Production and Average Amperage Data for Well H17.**

The weight factor represents the ratio of the calculated production of each well to the total calculated production, with the assumption that wells with higher calculated production contribute more to the total error. The production measurement result for each well after the total production split correction is:

$$Q_i = Q_{ci} + \omega_i \Delta Q \qquad (12)$$

For example, in the Tarim Oilfield, due to equipment constraints, wells 72H and 23-1H share the same production process, and only the total production volume of the two wells is metered. Using the production split algorithm mentioned above, the production volumes for these two wells were calculated, as shown in the figure below Fig 11. The production split results closely match the trends observed in the current values.

In practical applications, the performance of electrical submersible pump and motors may gradually decline over time, while the physical properties of wellbore fluids, such as viscosity and density, can dynamically change due to the production stage. To maintain the prediction accuracy of the production model, it is essential to establish a dynamic correction mechanism based on measured flow rate data. Notably, the dynamic variations in equipment power parameters exhibit significant nonlinear characteristics and cannot be adequately characterized by simple linear relationships. Therefore, frequent corrections should be performed based on field operating conditions. Through continuous data iteration and updates, the predictive results of the system model can remain highly consistent with actual operating conditions.

### 4.3 Fault diagnosis and early warning

The daily production volume of an oil well is an important parameter and has significant reference value for fault diagnosis and early warning of the well [28,29]. For example, comparing the daily production volume calculated by the model with the actual metered surface production volume can provide useful insights for diagnosing and warning of casing leaks: when a casing leak occurs, the production volume through the electric pump will not show significant changes, but the surface production volume will noticeably decrease. When a casing leak occurs, the model-calculated daily production volume will be higher than the actual metered surface production volume. For example, in Well 14H, the surface-metered production volume and the model-calculated production volume trend are shown in the following figure Fig 12.

From the figure, it can be seen that after January 4, 2024, the model-calculated production volume remained relatively stable, while the surface-metered production volume decreased significantly. Therefore, it was concluded that a casing leak might have occurred in this well. Subsequent operational results confirmed this assumption.

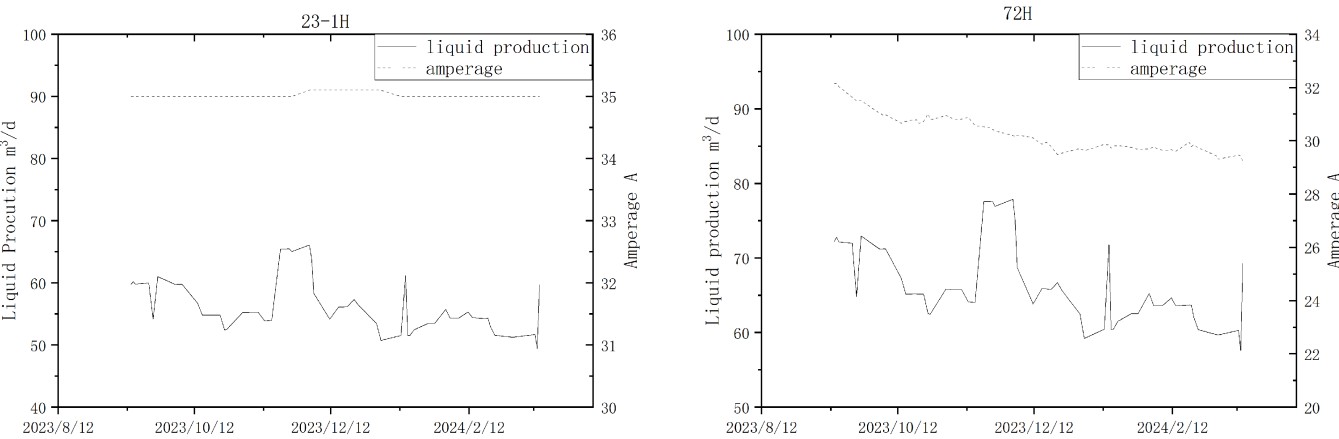

**Fig 11. Production Split Results for Wells 23-1H and 72H and Operating Current Trends.**

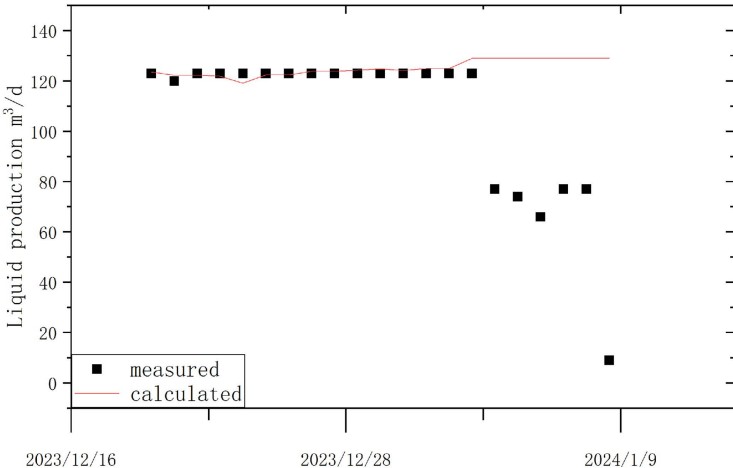

**Fig 12. Comparison Chart of Surface Metered Production and Model Calculated Production for Well 14H.**

## 5 Conclusion

(1) A digital production calculation method suitable for ESP wells without downhole pressure measurement devices is proposed in this paper. The method uses historical data such as surface casing pressure, dynamic liquid level, and production rate, along with the performance curves of the pump and motor, reservoir fluid properties, and real-time measurements of surface current and wellhead tubing pressure. It calculates the required input power for the pump and the actual output power of the motor, and through power balance, determines the real-time production rate of the well.

(2) The model takes into account the effects of equipment aging and additional power consumption by auxiliary devices, introducing the concept of correction power, which allows the model to be calibrated using metered production data. In addition, the model includes methods for correcting the pump performance curve under viscous liquid further improving the accuracy of the calculations.

(3) Compared to traditional manual metering, this model enables low-cost, high-frequency metering, significantly improving the accuracy and efficiency of production calculations. It allows for effective monitoring of dynamic changes in oilfield wells, promotes the intelligent management of oilfield pumping systems, and has significant implications for improving management efficiency and production benefits in oilfields.

(4) A production allocation algorithm for combined metering wells is proposed. By applying this algorithm, it is possible to calculate the actual production rate of each well based on the combined surface metered production. Additionally, by comparing the model-calculated production with the surface metered production, the algorithm can promptly detect well failures such as casing leakage.

## Supporting information

**S1. Table: The parameters involved in the calculation process of the model in this article, which are extracted from the daily production data sheets of Tarim Oilfield.** Due to the fact that specific data involves oilfield commercial secrets and internal management information, this table only displays key parameters after desensitization processing, and is for reference only.

## Author contributions

**Conceptualization:** Da Guo, Yingbin Liu, Guoxiang Zhang.

**Data curation:** Yongcang Ren.

**Funding acquisition:** Yongcang Ren, Yizhen Sun.

**Investigation:** Yizhen Sun.

**Methodology:** Yongcang Ren, Da Guo, Yizhen Sun, Bao Zhang, Jianxin Shen, Yinglin Zhang, Jinsong Yao.

**Project administration:** Yongcang Ren, Yinglin Zhang.

**Resources:** Yongcang Ren, Yingbin Liu, Yinglin Zhang, Guoxiang Zhang, Xueling Du.

**Software:** Guoxiang Zhang.

**Supervision:** Yongcang Ren, Yizhen Sun, Yingbin Liu, Yinglin Zhang, Xueling Du.

**Validation:** Yongcang Ren, Yizhen Sun, Yingbin Liu, Yinglin Zhang, Guoxiang Zhang.

**Visualization:** Yongcang Ren, Da Guo, Bao Zhang, Jianxin Shen, Yingbin Liu, Jinsong Yao.

**Writing – original draft:** Yongcang Ren, Da Guo, Bao Zhang, Jianxin Shen, Jinsong Yao.

**Writing – review & editing:** Yongcang Ren, Da Guo, Yizhen Sun.

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
