## [Decision Letter · Decision Letter 0]

7 Mar 2025

Dear Dr. Sun,

Thank you for submitting your manuscript to PLOS ONE. After careful consideration, we feel that it has merit but does not fully meet PLOS ONE’s publication criteria as it currently stands. Therefore, we invite you to submit a revised version of the manuscript that addresses the points raised during the review process.

We look forward to receiving your revised manuscript.

Kind regards,

Fahd Saeed Alakbari, Ph.D.

Academic Editor

PLOS ONE

Journal Requirements:

Reviewers' comments:

Reviewer's Responses to Questions

**Comments to the Author**

1. Is the manuscript technically sound, and do the data support the conclusions?

Reviewer #1: Yes

2. Has the statistical analysis been performed appropriately and rigorously?

Reviewer #1: N/A

3. Have the authors made all data underlying the findings in their manuscript fully available?

Reviewer #1: Yes

4. Is the manuscript presented in an intelligible fashion and written in standard English?

Reviewer #1: Yes

Reviewer #1: As the manuscript is novel, still some modification is required:

1 The description of the mathematical model could be slightly expanded to briefly outline key equations or assumptions.

2 It should be made clear how viscosity adjustment contributes to accuracy in the power calculation process.

3 The statement "calculated results align well with the metered results if periodically corrected" is vague. Consider specifying how frequently corrections are needed.

4 One crucial component of the model is how the pump performance curves are modified for viscous liquids. Give some thought to explaining the process by which these modifications were made and any restrictions on using them for certain fluid characteristics.

**Do you want your identity to be public for this peer review?** For information about this choice, including consent withdrawal, please see our Privacy Policy

Reviewer #1: No

---

## [Author Response · Author response to Decision Letter 1]

25 Mar 2025

Dear Editor and Reviewers,

We sincerely appreciate the time and effort you have dedicated to reviewing our manuscript. Your constructive comments have significantly improved the quality of our work. Below, we provide a point-by-point response to the reviewer’s suggestions, with corresponding revisions highlighted in the updated manuscript (v3-Revised Manuscript with Track Changes.docx).

Response to Journal Requirements:

1.Formatting and File Naming Compliance

We have strictly followed PLOS ONE's formatting templates and file naming requirements.

2.Code Availability Statement

This study does not involve any author-generated code, thus the code sharing policy is not applicable.

3.Reference Verification

We have thoroughly verified all references:

1.Ensured completeness (all citations match reference entries)

2.Confirmed no retracted papers were cited

3.Standardized formatting per PLOS ONE guidelines

Response to Reviewer’s Comments

Comment 1:The description of the mathematical model could be slightly expanded to briefly outline key equations or assumptions.

Response:

We have expanded the mathematical model description in Section 1 to explicitly clarify the key assumptions and equations:

1.Added the assumption that "the ESP well production system operates under steady-state conditions, where the torque of the pump and the rotational speed of the motor remain constant", ensuring the validity of the energy conservation principle.

2.Highlighted the revised equation (Equation 1) incorporating the correction power term , which accounts for energy losses due to auxiliary components and aging.

3.Clarified the definition of the numerical constant 11.574 in Equation 2.

4.Added supplementary explanations for the motor output power calculations.

Comment 2:It should be made clear how viscosity adjustment contributes to accuracy in the power calculation process.

Response

In Section 2.2, the specific reasons for improving power calculation accuracy through viscosity correction are added, and a new reference is added to elaborate on the mechanism of viscosity influence on pump efficiency.

Comment 3:The statement "calculated results align well with the metered results if periodically corrected" is vague. Consider specifying how frequently corrections are needed.

Response:

We acknowledge that correction power variations may not follow linear time-dependent patterns. Therefore, instead of specifying fixed intervals, we recommend performing corrections as frequently as field conditions permit. Accordingly:

1.Removed the term "periodically" from the Abstract to avoid ambiguity.

2.Added clarification in Section 4.1 (last paragraph).

Comment 4:One crucial component of the model is how the pump performance curves are modified for viscous liquids. Give some thought to explaining the process by which these modifications were made and any restrictions on using them for certain fluid characteristics.

Response:

We have made the following improvements:

1.Added a comprehensive viscosity correction flowchart (Figure 4) in Section 2.2 to visualize the step-by-step procedure.

2.Regarding restrictions on fluid viscosity correction,we recognize this as a valuable suggestion for future research�but the ANSI standard does not explicitly specify limitations, so we do not provide any additional explanations in the original text.

Conclusion

We believe these revisions address all reviewer concerns and strengthen the manuscript’s technical rigor. Thank you again for your valuable feedback. Please let us know if further modifications are needed.

Sincerely,

[Sun Yizhen]

[Research Institute of Petroleum Exploration and Development]

[Phone: +86 13263398197]

[Email: 605182894@qq.com]

---

## [Decision Letter · Decision Letter 1]

30 Jul 2025

Modeling and application of production metering for electric pump wells without downhole pressure measurement devices

PONE-D-25-01054R1

Dear Dr. Sun,

We’re pleased to inform you that your manuscript has been judged scientifically suitable for publication and will be formally accepted for publication once it meets all outstanding technical requirements.

Kind regards,

Sameer Sheshrao Gajghate, PhD

Academic Editor

PLOS ONE

Additional Editor Comments (optional):

Based on the reviewers' comments and initial check of the manuscript. The modified manuscript submitted by the author is provisionally accepted for possible publication in a journal.

Reviewers' comments:

Reviewer's Responses to Questions

**Comments to the Author**

Reviewer #1: All comments have been addressed

Reviewer #2: All comments have been addressed

2. Is the manuscript technically sound, and do the data support the conclusions?

Reviewer #1: Yes

Reviewer #2: Yes

3. Has the statistical analysis been performed appropriately and rigorously?

Reviewer #1: Yes

Reviewer #2: N/A

4. Have the authors made all data underlying the findings in their manuscript fully available?

Reviewer #1: Yes

Reviewer #2: (No Response)

5. Is the manuscript presented in an intelligible fashion and written in standard English?

Reviewer #1: Yes

Reviewer #2: Yes

Reviewer #1: All the points are very well reported. I appreciate your effort and no further comments are required in this manuscript.

Reviewer #2: (No Response)

**Do you want your identity to be public for this peer review?** For information about this choice, including consent withdrawal, please see our Privacy Policy

Reviewer #1: No

Reviewer #2: No

---

## [Editor Report · Acceptance letter]

PONE-D-25-01054R1

PLOS ONE

Dear Dr. Sun,

I'm pleased to inform you that your manuscript has been deemed suitable for publication in PLOS ONE. Congratulations! Your manuscript is now being handed over to our production team.

Kind regards,

on behalf of

Dr. Sameer Sheshrao Gajghate

Academic Editor

PLOS ONE